# Progesterone and prolactin levels in pregnant women living with HIV who delivered preterm and low birthweight infants: A nested case-control study

**Benjamin H. Chi**[1]*, **Dorothy Sebikari**[2], **Sean S. Brummel**[3], **Patricia DeMarrais**[3], **Rachel Chamanga**[4], **Maxensia Owor**[2], **Sufia Dadabhai**[5], **Joan T. Price**[1], **Taha Taha**[5], **Jeffrey Stringer**[1], **Mary Glenn Fowler**[6]

1 School of Medicine, University of North Carolina, Chapel Hill, North Carolina, United States of America, 2 Makerere University–Johns Hopkins University Research Collaboration, Kampala, Uganda, 3 Harvard T. H. Chan School of Public Health, Boston, Massachusetts, United States of America, 4 College of Medicine-Johns Hopkins Research Project, Blantyre, Malawi, 5 Johns Hopkins University Bloomberg School of Public Health, Baltimore, Maryland, United States of America, 6 Johns Hopkins University School of Medicine, Baltimore, Maryland, United States of America

* bchi@med.unc.edu

**Data Availability Statement:** Due to ethical restrictions in the study's informed consent documents and in the IMPAACT Network's

## Abstract

### Background

Antiretroviral therapy (ART) is associated with high rates of adverse birth outcomes, including preterm birth and low birthweight. Studies suggest that progesterone and prolactin may play important intermediary roles.

### Methods

We analyzed data from the Antenatal Component of the PROMISE trial, a multi-center study of pregnant women taking antiretroviral regimens (lopinavir/ritonavir-containing ART or zidovudine alone) to prevent mother-to-child HIV transmission. In a nested case-control study, we compared data from women who gave birth to preterm (<37 weeks gestation) and/or low birthweight (<2500 g) infants to matched individuals who did not. We measured serum progesterone and prolactin at 24–34 weeks gestation. We used conditional logistic regression to describe relationships between hormone levels, birth outcomes, and antiretroviral regimens.

### Results

299 women and their newborns were included (146 cases, 153 controls). When compared to women receiving zidovudine alone, those on ART had higher odds of progesterone levels under the 10[th] percentile (adjusted odds ratio [AOR]:2.34, 95%CI:1.41–3.89) and 25[th] percentile (AOR:2.07, 95%CI:1.46–2.94). However, higher levels of progesterone—rather than lower levels—were associated with our composite case outcome at the 10[th] percentile

approved human subjects protection plan, study data are available upon request from sdac.data@sdac.harvard.edu with the written agreement of the International Maternal Pediatric Adolescent AIDS Clinical Trials (IMPAACT) network. Data are also available to all interested researchers upon request to the IMPAACT Statistical and Data Management Center's data access committee (email address: sdac.data@fstrf.org); this committee reviews and responds to requests for data, obtains necessary approvals from IMPAACT leadership and the NIH, arranges for signature of a Data Use Agreement, and sends the requested data.

**Funding:** Overall support for the International Maternal Pediatric Adolescent AIDS Clinical Trials Network (IMPAACT) was provided by the National Institute of Allergy and Infectious Diseases (NIAID) with co-funding from the Eunice Kennedy Shriver National Institute of Child Health and Human Development (NICHD) and the National Institute of Mental Health (NIMH), all components of the National Institutes of Health, under Award Numbers UM1AI068632 (IMPAACT LOC), UM1AI068616 (IMPAACT SDMC) and UM1AI106716 (IMPAACT LC), and by NICHD contract number HHSN275201800001I. Additional investigator support was provided by NIAID (BHC, K24AI120796) and the Fogarty International Center (JTP, K01TW010857). The study products in the PROMISE trial were provided free of charge by Abbott, Gilead Sciences, Boehringer Ingelheim, and GlaxoSmithKline. The content is solely the responsibility of the authors and does not necessarily represent the official views of the National Institutes of Health. For this substudy, the funders had no role in study design, data collection and analysis, decision to publish, or preparation of the manuscript.

**Competing interests:** The study products in the PROMISE trial were provided free of charge by Abbott, Gilead Sciences, Boehringer Ingelheim, and GlaxoSmithKline. Funders had no role in study design, data collection and analysis, decision to publish, or preparation of the manuscript. This does not alter our adherence to PLOS ONE policies on sharing data and materials.

(AOR:1.88, 95%CI:0.77–4.59) and 25th percentile (AOR:1.96, 95%CI:1.06–3.61). Associations were not observed between prolactin, antiretroviral regimen, and birth outcomes.

## Conclusion

We observed lower progesterone levels among women allocated to ART regimens; however, higher progesterone levels were associated with preterm birth and/or low birthweight. While features of the study design may have contributed to these findings, they nevertheless highlight the potentially complex mechanisms underpinning adverse birth outcomes and HIV.

## Introduction

Three-drug, combination antiretroviral therapy (ART) has been shown to dramatically reduce vertical HIV transmission to less than 2%, including in breastfeeding populations [1–4]. Universal provision of ART to pregnant and breastfeeding women living with HIV has led to major reductions in the global pediatric HIV burden and is considered a cornerstone in worldwide efforts to eliminate mother-to-child transmission of HIV [5]. However, ART may present important side effects during pregnancy as well. Its use has been associated with adverse birth outcomes, including preterm birth and low birth weight, a risk that may be greater when treatment is initiated prior to conception [6,7]. Pregnancy outcomes such as preterm birth can result in significant morbidity and mortality [8], particularly in settings where HIV prevalence is high and resources are limited [9].

The association between ART and preterm birth has been prominent among the drug class of protease inhibitors (PIs), which includes lopinavir, atazanavir, and ritonavir [10–12]. Although the World Health Organization now categorizes several PIs as second-line agents [13], studying their use during pregnancy may provide broader insights into the relationship between ART and adverse birth outcomes. For example, PIs have been associated with placental vascular changes that may in turn lead to fetal growth restriction [14]. The use of lopinavir/ritonavir (LPV/r)-containing ART regimens has been associated with elevated estradiol levels during pregnancy, which also could negatively affect fetal growth [15,16]. Mouse pregnancy models have shown that PI-containing ART may be associated with decreased plasma progesterone, which in turn is correlated with decreased fetal and placental weight. Interestingly, this effect could be reversed—at least partially—via progesterone supplementation [17]. When compared to HIV-negative controls, plasma progesterone was also lower among women living with HIV on PI-containing regimens and this appeared to be modulated by lower circulating prolactin [18].

In the Antenatal Component of the multi-country Promoting Maternal and Infant Survival Everywhere (PROMISE) trial, 3,529 pregnant women living with HIV were allocated to receive an LPV/r-containing ART regimen (with either zidovudine/lamivudine [ZDV/3TC] or tenofovir/emtricitabine [TDF/FTC]) versus zidovudine prophylaxis alone over their antenatal course. Women randomized to either ART regimen had lower vertical HIV transmission rates compared to those on zidovudine prophylaxis (0.5% vs. 1.8%); however, they also experienced higher rates of adverse birth outcomes [4]. Compared to the control arm, women on ZDV-based ART (adjusted odds ratio [AOR]: 1.82, 95% confidence interval [CI]: 1.47–2.26) and TDF-based ART (AOR: 1.77, 95%CI: 1.29–2.43) had greater risk for preterm birth under 37 weeks. Similar findings were noted among low birthweight infants as well [4,19]. To

understand the causes underlying this phenomenon, we conducted a case-control study nested within the larger PROMISE cohort. Specifically, we investigated the association between adverse birth outcomes (i.e., preterm birth or low birth weight) and mid-pregnancy measurements of progesterone and prolactin.

## Methods

### Study design and outcomes

The Antenatal Component of PROMISE 1077BF/FF was designed to compare the efficacy of zidovudine prophylaxis vs. ART taken during pregnancy (NCT01061151 and NCT01253538). The methods of the parent trial have been described in depth elsewhere [4]. Briefly, HIV-positive pregnant women who did not meet local clinical or immunologic criteria for ART were eligible for enrollment from 14 weeks gestation onward. Women were randomized to receive one of three regimens: antenatal twice daily ZDV, along with intrapartum nevirapine and a seven-day "tail" of emtricitabine/tenofovir (i.e., antenatal ZDV prophylaxis); antenatal ART comprising ZDV, 3TC, and LPV/r (i.e., ZDV-based ART); or antenatal ART comprising TDF, FTC, and LPV/r (i.e., TDF-based ART). In early versions of the protocol, women who screened negative for hepatitis B surface antigen (HBsAg) were randomized only to antenatal ZDV prophylaxis or ZDV-based ART, while women who screened positive for HBsAg were randomized to one of three arms. Based on emerging data about the safety of TDF in pregnancy, beginning in Version 3.0 of the protocol (August 2012), all women were allocated with equal probability to the three antiretroviral regimens.

Using data and specimens collected within the PROMISE study, we conducted a nested case-control study to investigate the associations between mid-pregnancy progesterone and prolactin levels, and adverse birth outcomes. Delivery and early neonatal outcomes were collected on all infants born in the study. Because routine obstetrical ultrasound was unavailable at most study sites, gestational age at birth was primarily determined by the pediatrician's newborn examination (i.e., New Ballard Score). When such data were not available, gestational age at birth was determined by—in order of priority—the obstetrician's estimate during labor, other pregnancy outcome information, or calculated by the initial antenatal assessment of expected delivery date [20].

### Study participants

This substudy analyzed stored specimens and clinical data from two major enrollment sites for the PROMISE study: Makerere University–Johns Hopkins University Research Collaboration (Kampala, Uganda) and College of Medicine-Johns Hopkins Research Project (Blantyre, Malawi). Eligibility criteria included: maternal consent for non-protocol-specified use of stored specimens, available plasma specimen between 24 and 34 weeks of pregnancy, documented antiretroviral regimen start date, and birth of liveborn infant with sufficient data for classification as either a case or control.

Cases were defined as mother-infant pairs in which a singleton newborn was born at less than 37 weeks (preterm birth, or PTB) and/or weighed less than 2500 g at birth (low birth-weight, or LBW). This composite definition was used to address potential misclassification associated with New Ballard Score [21]. All participants from our two target sites who met these criteria were included in the analysis. Controls were defined as mother-infant pairs in which the newborn was born at 37 weeks or greater and weighed at least 2500 g at birth. Both of these measurements were required in order to be considered for the control group. Controls were selected at a 1:1 ratio with cases and matched according to country, infant sex, and gestational age when the mother started antiretroviral agents (categorized as <20 weeks, 20 to <28

weeks, and ≥28 weeks gestational age). They were then individually matched according to randomization date. Prior to study activation at each site, ethical approvals were obtained by local research ethics committees and partnering US institutional review boards.

All participants had at least one stored maternal plasma specimen between 24 and 34 weeks of gestation, calculated according to the gestational age at delivery. Because progesterone and prolactin increase steadily over the course of pregnancy [22,23], we sought to standardize the timing of measurement in the study population. We originally targeted the early third trimester (i.e., 28 to 32 weeks gestational age), reasoning that it was proximal to the time of delivery yet early enough for preventive interventions. However, due to the distribution of specimen collection (which was timed from study entry rather than specific gestational ages), we expanded the eligibility window and standardized results based on their timing within that window (see below). Nevertheless, when more than one specimen was available within this window, the one closest to 32 weeks gestational age was analyzed. Each specimen was analyzed for progesterone and prolactin using the Cobas® platform (Roche Diagnostics, Indianapolis, IN, USA).

## Statistical analysis

For each participant, we measured progesterone and prolactin at a single time point. In our analysis, we described the distribution of these measurements, using locally estimated scatterplot smoothing (LOESS) to illustrate trends over time for the study population. Because hormonal levels change over the course of pregnancy, we also conducted stratified comparisons across gestational age categories: 24 to <26 weeks, 26 to <28 weeks, 28 to <30 weeks, 30 to <32 weeks, and 32 to 34 weeks.

Using logistic regression, we separately examined the association between antiretroviral regimen and progesterone and prolactin levels. Individuals randomized to LPV/r-containing combination regimens, regardless of nucleoside reverse transcriptase inhibitor backbone, were combined into a single ART exposure group. These were compared to participants randomized to the ZDV-only prophylaxis exposure group. Because there are no established thresholds associated with adverse birth outcomes, we explored the relationship between ART regimen and hormone levels at the 10th and 25th percentiles. These thresholds were calculated within each of the gestational age strata described above. We adjusted for gestational age using a b-spline model. To determine associations between antiretroviral regimens and hormone levels, weights computed according to the case/control matching criteria were also employed to effectively up-weight the controls and better replicate the full PROMISE study population. This was necessary because of the under-sampling of controls through our matching process.

We used conditional logistic regression to estimate the odds of our composite PTB-LBW outcome at different progesterone and prolactin levels. Again, we used different thresholds for progesterone and prolactin to better understand the potential association with PTB-LBW, this time at the 10th and 25th percentiles only. Results were stratified by infant sex, country, and gestational age at start of antiretroviral regimen. They were also adjusted for antiretroviral regimen, CD4 at screening, HIV RNA at baseline, pregnancy history, smoking history, alcohol history, age, body mass index, and year of randomization.

We conducted sensitivity analyses using PTB alone as an outcome. Although part of our primary composite outcome, LBW actually comprises three different conditions: preterm infants, growth-restricted term infants, and constitutionally small term infants. We reasoned that excluding those cases who were LBW only in such a sensitivity analysis could provide further insight into the relationships between adverse birth outcomes, antiretroviral regimens, and hormonal (i.e., progesterone and prolactin) levels. The clinical data used in this analysis

was downloaded and frozen as of May 1, 2018. P values of <0.05 were considered statistically significant. All analyses were performed using SAS version 9.4 (Cary, NC, USA).

### Ethics statement

The PROMISE 1077BF/FF was approved at the following participating institutions: Johns Hopkins School of Medicine Institutional Review Board (U.S.; NA_00041835, NA_0003999), College of Medicine Research and Ethics Committee (Malawi; P.05/10/950), and Joint Clinical Research Centre Institutional Review Board (Uganda, no IRB number provided). All participating pregnant women provided written informed consent prior to enrollment. This included permission to use stored specimens for non-protocol-specified research purposes.

## Results

Between April 2011 and October 2014, 1072 HIV-positive pregnant women were enrolled into the PROMISE trial at our two target sites. Overall, 1004 women delivered a singleton liveborn infant and, of these, 609 women had a stored plasma specimen available between 24 and 34 weeks gestation. One hundred and fifty-three mother-infant pairs met criteria as cases and all were included in the study population. Of the remaining participants, 456 met eligibility criteria for the control group, from which 153 matched participants were selected based on characteristics described previously.

Among the cases, seven had incorrect specimens tested and were thus excluded from the full analysis. A comparison of the cases and controls is shown in Table 1. These include maternal differences in randomized antiretroviral regimen and baseline HIV RNA levels. Case infants were less likely to have gestational age determined at birth, but more likely to be of shorter length and have lower APGAR scores at birth.

Concentrations for progesterone and prolactin are shown continuously across the range of gestational ages at time of specimen collection (Fig 1). The superimposed LOESS lines suggest that progesterone levels were generally higher in the cases and lower in the controls during the collection period (i.e., 24–34 weeks gestation). In contrast, only marginal differences were noted for prolactin levels. Histograms depicting progesterone and prolactin distributions are also shown in S1 and S2 Figs.

We compared the relative distributions of progesterone and prolactin across individual two-week specimen collection windows, stratified by case/control status and antiretroviral regimens (Fig 2). Regardless of whether comparisons were made within the ZDV only or ART groups, during the second and early third trimesters, the cases consistently trended towards higher serum progesterone levels compared to controls. For prolactin, similar trends were noted in two gestational age strata (i.e., 26 to <28 weeks, 28 to <30 weeks); however, these differences largely disappeared as gestation progressed.

Compared to receipt of ZDV during pregnancy, antenatal ART was consistently and significantly associated with low progesterone levels when defined as under the 10th percentile (adjusted odds ratio [AOR]: 2.34, 95% confidence interval [CI]: 1.41, 3.89) and under the 25th percentile (AOR: 2.07, 95%CI: 1.46, 2.94). Such trends were inconsistently observed with low prolactin levels and none were statistically significant (Table 2).

We examined the relationship between hormone levels and the PTB-LBW outcome (Table 3). Higher maternal progesterone levels at 24–34 weeks gestation were associated with the PTB-LBW outcome for both ART and ZDV regimens. In adjusted analyses, these trends were similar whether a ≥10th percentile threshold (AOR: 1.88, 95%CI: 0.77, 4.59) or ≥25th percentile threshold (AOR: 1.96, 95%CI: 1.06, 3.61) was considered. Although similar trends were observed for prolactin, these did not meet our definition of statistical significance.

**Table 1. Baseline maternal and infant characteristics for mother-infant pairs included in this analysis.**

| | Case (N = 146) | Control (N = 153) |
|---|---|---|
| **Maternal characteristics** | | |
| Age at randomization in years, median (Q1, Q3) | 25.9 (22.4, 29.4) | 25.7 (22.5, 29.5) |
| Site, n (%) | | |
| Kampala, Uganda | 38 (26%) | 41 (27%) |
| Blantyre, Malawi | 108 (74%) | 112 (73%) |
| Randomization arm—antenatal antiretroviral regimen, n (%) | | |
| Lopinavir/ritonavir + zidovudine + lamivudine | 81 (55%) | 52 (34%) |
| Lopinavir/ritonavir + tenofovir + emtricitabine | 20 (14%) | 17 (11%) |
| Zidovudine only | 45 (31%) | 84 (55%) |
| Basis of gestational age dating, n (%) | | |
| Prenatal obstetric evaluation | 3 (2%) | 2 (1%) |
| Intrapartum obstetric evaluation | 5 (3%) | – |
| Newborn examination | 130 (89%) | 151 (99%) |
| Postnatal determination | 8 (5%) | – |
| Gestational age at regimen initiation in weeks, median (Q1, Q3) | 23.6 (20.4, 26.6) | 24.1 (20.4, 26.9) |
| Gestational age at specimen collection in weeks, median (Q1, Q3) | 30.7 (28.4, 32.4) | 30.9 (28.6, 32.1) |
| CD4 at screening, cells/mm$^3$, median (Q1, Q3) | 514 (434, 656) | 514 (420, 637) |
| World Health Organization clinical staging at screening, n (%) | | |
| Stage 1 | 142 (97%) | 149 (97%) |
| Stage 2 | 4 (3%) | 4 (3%) |
| HIV RNA level prior to randomization, median (Q1, Q3) | 12,537 (2,985, 54,091) | 9,255 (2,509, 34,937) |
| HIV RNA level prior to randomization, n (%) | | |
| Missing | 1 (<1%) | – |
| Below lower limit of quantification | 6 (4%) | 3 (2%) |
| < 400 copies/mL | 5 (3%) | 8 (5%) |
| 400 to <1,000 copies/mL | 7 (5%) | 10 (7%) |
| 1,000 to <10,000 copies/mL | 47 (32%) | 57 (37%) |
| 10,000 to <100,000 copies/mL | 61 (42%) | 65 (42%) |
| ≥ 100,000 copies/mL | 19 (13%) | 10 (7%) |
| **Infant characteristics** | | |
| Infant sex, n (%) | | |
| Female | 79 (54%) | 81 (53%) |
| Male | 67 (46%) | 72 (47%) |
| Gestational age at birth, median (Q1, Q3) | 36 (34, 36) | 38 (38, 40) |
| Gestational age at birth, n (%) | | |
| < 28 weeks | 1 (<1%) | – |
| 28 to <34 weeks | 14 (10%) | – |
| 24 to <37 weeks | 102 (70%) | – |
| ≥ 37 weeks | 29 (20%) | 153 (100%) |
| Weight within five days of life (week 0 visit) in grams, median (Q1, Q3) | 2400 (2200, 2700) | 3000 (2800, 3300) |
| Weight within five days of life (week 0 visit) in grams, n (%) | | |
| Missing | 7 (5%) | – |
| < 2500 grams | 79 (54%) | – |
| ≥ 2500 grams | 60 (41%) | 153 (100%) |
| Eligibility criteria to be included as a case, n (%) | | |

(*Continued*)

**Table 1.** (Continued)

| | Case (N = 146) | Control (N = 153) |
|---|---|---|
| < 37 weeks gestation *and* <2500 grams at birth | 50 (34%) | – |
| < 37 weeks at birth only | 67 (46%) | – |
| < 2500 grams at birth only | 29 (20%) | – |

When analyses were stratified according to antiretroviral regimen (i.e., ART vs. ZDV alone), overall trends remained consistent. Among those women on ART during pregnancy, for example, progesterone levels that were ≥10th percentile (AOR: 1.62, 95%CI: 0.57, 4.56) and ≥25th percentile (AOR: 3.09, 95%CI: 1.44, 6.65) were associated with elevated odds for PTB or LBW. Among those on antenatal ZDV only, similar findings were observed at the ≥10th percentile threshold (AOR: 2.49, 95%CI: 0.45, 13.8) and ≥25th percentile (AOR: 2.13, 95%CI: 0.75, 6.02); however, in the latter sub-analysis, there was greater uncertainty in the estimates, likely due to the smaller sample size. Stratified analyses for prolactin by antiretroviral regimen allocation were consistent with our main findings (S1 Table).

These results were also consistent with sensitivity analyses that considered only PTB as the primary outcome (Table 3). The magnitude of association appeared to increase when progesterone was considered at the ≥10th percentile threshold (AOR: 4.17, 95%CI: 1.54, 14.4) and the ≥25th percentile threshold (AOR: 2.89, 95%CI: 1.43, 5.83). Again, the associations between prolactin and PTB appeared elevated at the ≥10th and ≥25th percentiles, but did not reach statistical significance.

## Discussion

We hypothesized that the higher PTB and LBW outcomes observed in the PROMISE trial among women taking PI-containing ART would be associated with lower levels of progesterone and prolactin. Our findings about progesterone, ART, and adverse birth outcomes did not support these hypotheses. The parent PROMISE study demonstrated a significant association between PI-containing ART and delivery outcomes such as PTB and LBW, consistent with other literature [4,19]. We also found that ART regimens were associated with lower progesterone levels, consistent with other studies [17,18]. However, the composite outcome of PTB and LBW was associated with *higher* progesterone levels, an association that appeared to strengthen when the endpoint was PTB alone.

This constellation of findings is difficult to interpret, but several factors may contribute. First, these results may relate to the measurement of gestational age in PROMISE. In a meta-analysis, Lee and colleagues found that Ballard score tended to overestimate gestational age by 0.4 weeks and dated 95% of infants within ±3.8 weeks of ultrasound dating [21]. In a subset of 720 participants with documented obstetric ultrasound in the PROMISE trial, performance of the New Ballard Score appeared to differ according to the definition threshold for PTB [20]. If this misclassification were more likely to occur in cases or the controls, a plausible scenario since these were defined by gestational age (or proxies thereof), this could affect our results. Since the estimated timing of specimen collection was back-calculated from the gestational age at delivery, misclassifications by the New Ballard Score would affect not only the outcome variable, but the exposure variables as well. Second, our definition for cases may have been insufficiently narrow. In previous studies, low progesterone during pregnancy has been associated with poor fetal growth [17]. While PTB and LBW are related conditions, they are not exactly the same. It is also possible that our window for specimen collection (i.e., 24 to 34 weeks) was

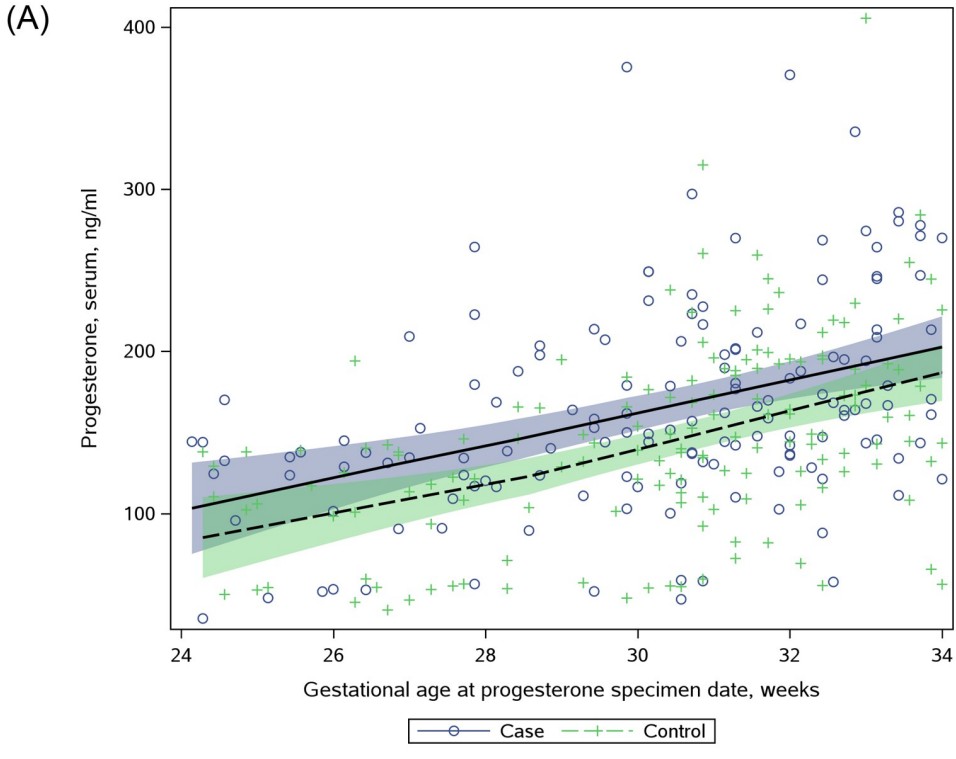

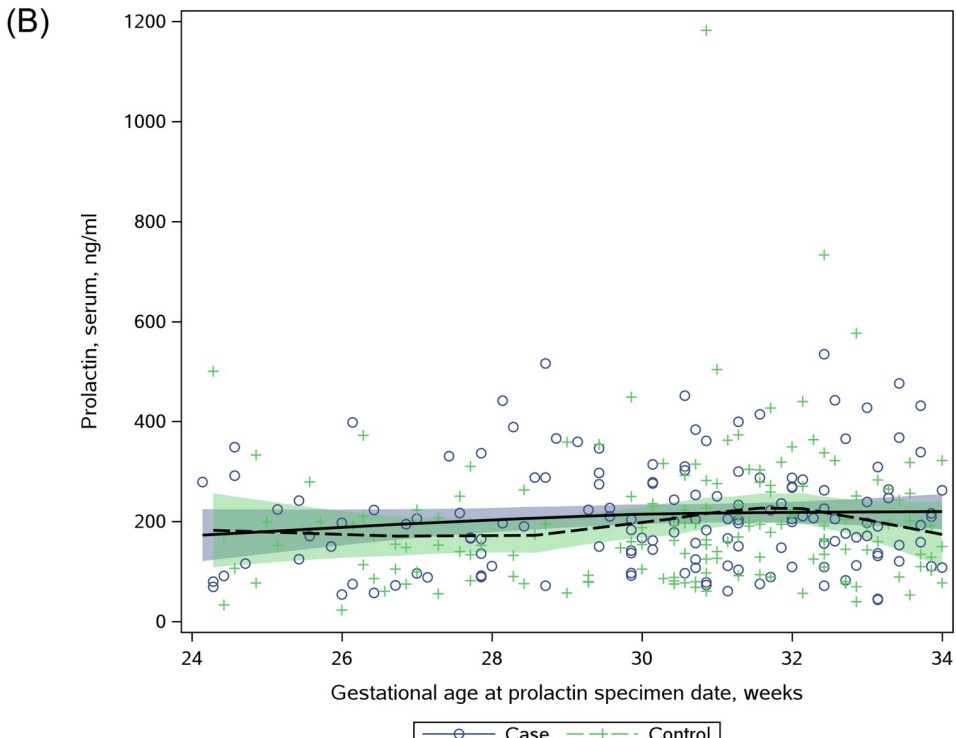

**Fig 1. Concentrations of progesterone (A) and prolactin (B) by gestational age at specimen collection.** Cases (+) and controls (o) are noted separately.

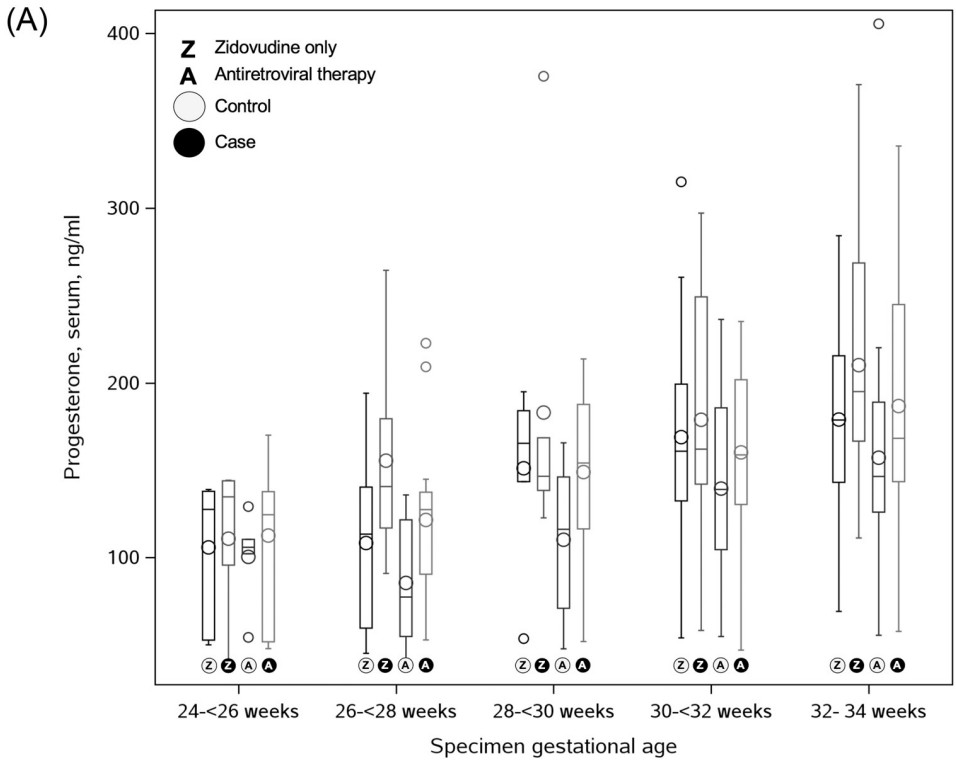

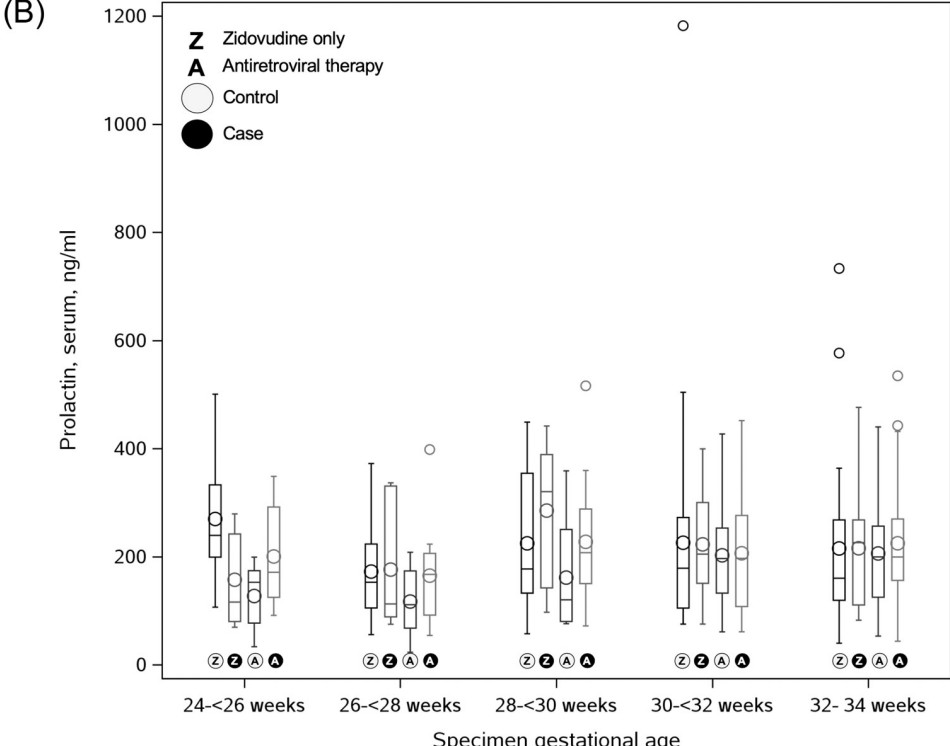

**Fig 2. Distribution of progesterone (A) and prolactin (B) by antenatal antiretroviral regimen and gestational age at specimen collection.** ZDV = zidovudine, ART = antiretroviral therapy.

**Table 2. Relationship of antiretroviral regimen arm and odds of low hormone levels.**

| Threshold* | | Unadjusted odds ratio (95%CI) | p value | Adjusted** odds ratio (95%CI) | p value |
|---|---|---|---|---|---|
| **Progesterone** | | | | | |
| < 10th percentile | ZDV only | Ref | – | Ref | – |
| | LPV/r-based ART | 2.84 (1.76, 4.56) | <0.0001 | 2.34 (1.41, 3.89) | 0.001 |
| < 25th percentile | ZDV only | Ref | – | Ref | – |
| | LPV/r-based ART | 2.30 (1.64, 3.22) | <0.0001 | 2.07 (1.46, 2.94) | <0.0001 |
| **Prolactin** | | | | | |
| < 10th percentile | ZDV only | Ref | – | Ref | – |
| | LPV/r-based ART | 1.13 (0.72, 1.76) | 0.59 | 1.28 (0.77, 2.14) | 0.33 |
| < 25th percentile | ZDV only | Ref | – | Ref | – |
| | LPV/r-based ART | 1.08 (0.77, 1.51) | 0.66 | 1.12 (0.78, 1.62) | 0.54 |

* Percentiles based on levels within corresponding gestational age strata.

** Adjusted models were weighted by case-control status, infant sex, country, and gestational age at start of antenatal antiretroviral regimen.

ZDV = zidovudine. LPV/r = lopinavir/ritonavir. ART = antiretroviral therapy.

too wide or mis-timed over the course of pregnancy. Progesterone levels later in pregnancy, for example, might better indicate fetal weight gain in the final trimester. Third, there are numerous illustrations of how bias may explain seemingly paradoxical findings [24]. Once bias is introduced in the design phase—regardless of the type or its source—it can be difficult to reconcile in the analysis stage.

**Table 3. Relationship of hormone level and odds of adverse birth outcomes.**

| | Threshold* | Unadjusted odds ratio (95%CI) | p value | Adjusted** odds ratio (95%CI) | p value |
|---|---|---|---|---|---|
| **Preterm birth or low birth weight** | | | | | |
| Progesterone | < 10th percentile | Ref | – | Ref | – |
| | ≥ 10th percentile | 1.81 (0.80, 4.06) | 0.153 | 1.88 (0.77, 4.59) | 0.166 |
| | < 25th percentile | Ref | – | Ref | – |
| | ≥ 25th percentile | 1.71 (0.99, 2.94) | 0.055 | 1.96 (1.06, 3.61) | 0.031 |
| Prolactin | < 10th percentile | Ref | – | Ref | – |
| | ≥ 10th percentile | 1.51 (0.69, 3.33) | 0.302 | 1.58 (0.65, 3.81) | 0.311 |
| | < 25th percentile | Ref | – | Ref | – |
| | ≥ 25th percentile | 1.36 (0.80, 2.32) | 0.261 | 1.30 (0.71, 2.40) | 0.397 |
| **Preterm birth only** | | | | | |
| Progesterone | < 10th percentile | Ref | – | Ref | – |
| | ≥ 10th percentile | 3.66 (1.28, 10.40) | 0.015 | 4.71 (1.54, 14.4) | 0.007 |
| | < 25th percentile | Ref | – | Ref | – |
| | ≥ 25th percentile | 2.16 (1.16, 4.01) | 0.015 | 2.89 (1.43, 5.83) | 0.003 |
| Prolactin | < 10th percentile | Ref | – | Ref | – |
| | ≥ 10th percentile | 1.70 (0.69, 4.17) | 0.246 | 1.65 (0.62, 4.39) | 0.314 |
| | < 25th percentile | Ref | – | Ref | – |
| | ≥ 25th percentile | 1.94 (1.06, 3.57) | 0.033 | 1.81 (0.92, 3.54) | 0.085 |

* Percentiles based on levels within corresponding gestational age strata.

** Stratified by infant sex, country, and gestational age at start of antiretroviral regimen. Further adjusted for antiretroviral regimen, CD4 at screening, HIV RNA at baseline, pregnancy history, smoking history, alcohol history, age, body mass index, and year of randomization.

The relationships between ART, progesterone, and adverse birth outcomes may also be more complex than originally hypothesized. They may not be causal, but instead work through other intermediary pathways. For example, the changes in progesterone and estradiol during pregnancy are well-documented. In their comparison of LPV/r- vs. efavirenz-containing ART regimens, McDonald, et al. found that LPV/r exposure was associated with an increase in estradiol—while efavirenz was associated with a decrease in estradiol—when compared to an ART-naïve gestational age-matched comparator group (p<0.001 for both comparisons). Despite their opposite directionality, these derangements in estradiol were both associated with adverse birth outcomes. Higher estradiol levels in the LPV/r arm were associated with small for gestational age (SGA) infants (p = 0.027). Lower estradiol levels in the efavirenz arm correlated with SGA (p = 0.0019) and LBW (p = 0.019). Progesterone levels did not differ among participants on either ART regimen, though lower levels were associated with SGA in the LPV/r arm alone (p = 0.04) [15].

For women on PI-containing ART, prolactin may play a role in these decreased progesterone levels. Papp, et al. showed that placental expression of the progesterone inactivating enzyme, 20-alpha-hydroxysteriod dehydrogenase (20α-HSD), was elevated among pregnant women on PI-containing ART, which could in turn lead to lower maternal progesterone levels. Prolactin is known to suppress 20α-HSD and when prolactin levels are low—as was observed by Papp and colleagues—it may contribute to preterm birth [18]. We investigated whether prolactin levels were associated with adverse birth outcomes. In our analysis, however, prolactin levels did not differ between cases and controls.

Analyzing data from the large, multi-center PROMISE trial, we sought to gain further insights about ART-associated preterm birth. Our case-control design leveraged the parent study's randomization scheme and in-built comparison groups. Despite these strengths, we note several limitations. First, this nested study was conducted in only two sites for logistical reasons. However, efforts were made to standardize participant follow-up in the parent study, including clinical and obstetrical management. These populations are also likely representative of other African settings, especially for the biological exposures of interest (i.e., progesterone, prolactin). Second, the relationship between ART and adverse birth outcomes is complex and the causal pathways poorly elucidated. The parent trial was not designed to investigate these potential causal relationships and, as a result, many relevant data elements were not collected. As a result, unmeasured confounding could play an important contributing role in seemingly paradoxical results. Third, the composition of the ART regimen—both individual agents and combination of agents—may play a crucial role in determining these mechanisms. In the IMPAACT 2010 trial, for example, the rates of preterm birth varied significantly by regimen: 5.8% among women randomized to dolutegravir (DTG), FTC, and tenofovir alafenamide; 9.4% among women randomized to DTG, FTC, and TDF; and 12.1% among those on efavirenz, FTC, and TDF [25]. Extension of our work to other antiretroviral regimens, especially those with lower preterm birth rates, may shed further light on this complex process.

In summary, in this nested case-control study within the PROMISE trial, we observed lower progesterone levels mid-pregnancy among women allocated to LPV/r-containing ART regimens. Similar to other reports, antenatal ART was also associated with higher odds of PTB or LBW compared to antenatal ZDV alone. However, regardless of antiretroviral regimen, PTB and LBW were associated with higher (rather than lower) progesterone, a finding that appears incongruous with prior studies on preterm birth. These latter findings were unexpected and may be an artifact of the study design. These results also highlight the complex mechanisms underlying PTB among HIV-positive pregnant women. The benefits of ART during pregnancy and breastfeeding are unquestioned, including reductions in vertical HIV transmission and improvements in maternal health. Further study is needed to understand specific

adverse events—including birth outcomes—and fully optimize regimens for mothers and their newborns.

## Supporting information

**S1 Fig. Histogram of progesterone concentrations across all participants.**
(PDF)

**S2 Fig. Histogram of prolactin concentrations across all participants.**
(PDF)

**S1 Table. Relationship of hormone level and odds of preterm birth or low birth weight, stratified by antenatal antiretroviral regimen.**
(PDF)

## Acknowledgments

We thank our study participants and study personnel for their contributions to the conduct of the parent trial. Testing for progesterone and prolactin was conducted at the laboratory of Dr. William Clarke at Johns Hopkins University School of Medicine.

## Author Contributions

**Conceptualization:** Benjamin H. Chi, Dorothy Sebikari, Sean S. Brummel, Mary Glenn Fowler.

**Data curation:** Dorothy Sebikari, Rachel Chamanga, Maxensia Owor, Sufia Dadabhai.

**Formal analysis:** Sean S. Brummel, Patricia DeMarrais.

**Funding acquisition:** Mary Glenn Fowler.

**Investigation:** Benjamin H. Chi, Dorothy Sebikari, Rachel Chamanga, Maxensia Owor, Sufia Dadabhai, Joan T. Price, Taha Taha, Jeffrey Stringer, Mary Glenn Fowler.

**Methodology:** Benjamin H. Chi, Sean S. Brummel, Patricia DeMarrais, Joan T. Price, Jeffrey Stringer.

**Project administration:** Taha Taha, Mary Glenn Fowler.

**Validation:** Sean S. Brummel, Patricia DeMarrais.

**Writing – original draft:** Benjamin H. Chi.

**Writing – review & editing:** Benjamin H. Chi, Dorothy Sebikari, Sean S. Brummel, Patricia DeMarrais, Rachel Chamanga, Maxensia Owor, Sufia Dadabhai, Joan T. Price, Taha Taha, Jeffrey Stringer, Mary Glenn Fowler.

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
