## [Decision Letter · Decision Letter 0]

12 May 2021

PONE-D-20-39706

Progesterone and prolactin levels in pregnant women living with HIV who delivered preterm and low birthweight infants: a nested case-control study

PLOS ONE

Dear Dr. Chi,

Thank you for submitting your manuscript to PLOS ONE. After careful consideration, we feel that it has merit but does not fully meet PLOS ONE’s publication criteria as it currently stands. Therefore, we invite you to submit a revised version of the manuscript that addresses the points raised during the review process.

We look forward to receiving your revised manuscript.

Kind regards,

Jianhong Zhou

Associate Editor

PLOS ONE

Journal Requirements:

2. In your Methods section, please provide additional information about the participant recruitment method and the demographic details of your participants. Please ensure you have provided sufficient details to replicate the analyses such as: a) a description of how participants were recruited, and b) descriptions of where participants were recruited and where the research took place (hospital/site name).

3. Please ensure you have included the registration number for the clinical trial referenced in the manuscript.

4. Please ensure you have discussed any potential limitations of your study in the Discussion, including study design, sample size and/or potential confounders.

5. Please provide additional details regarding participant consent. In the ethics statement in the Methods and online submission information, please ensure that you have specified (1) whether consent was informed and (2) what type you obtained (for instance, written or verbal, and if verbal, how it was documented and witnessed). If your study included minors, state whether you obtained consent from parents or guardians. If the need for consent was waived by the ethics committee, please include this information.

6. We note that you have indicated that data from this study are available upon request. PLOS only allows data to be available upon request if there are legal or ethical restrictions on sharing data publicly. For information on unacceptable data access restrictions, please see http://journals.plos.org/plosone/s/data-availability#loc-unacceptable-data-access-restrictions.

7. Thank you for stating the following in the Financial Disclosure section:

[Overall support for the International Maternal Pediatric Adolescent AIDS Clinical Trials Network (IMPAACT) was provided by the National Institute of Allergy and Infectious Diseases (NIAID) with co-funding from the Eunice Kennedy Shriver National Institute of Child Health and Human Development (NICHD) and the National Institute of Mental Health (NIMH), all components of the National Institutes of Health, under Award Numbers UM1AI068632 (IMPAACT LOC), UM1AI068616 (IMPAACT SDMC) and UM1AI106716 (IMPAACT LC), and by NICHD contract number HHSN275201800001I. Additional investigator support was provided by NIAID (BHC, K24AI120796) and the Fogarty International Center (JTP, K01TW010857). The study products in the PROMISE trial were provided free of charge by Abbott, Gilead Sciences, Boehringer Ingelheim, and GlaxoSmithKline. The content is solely the responsibility of the authors and does not necessarily represent the official views of the National Institutes of Health. For this substudy, the funders had no role in study design, data collection and analysis, decision to publish,  or preparation of the manuscript.]. 

We note that you received funding from a commercial source: Abbott, Gilead Sciences, Boehringer Ingelheim, and GlaxoSmithKline

Reviewers' comments:

Reviewer's Responses to Questions

**Comments to the Author**

1. Is the manuscript technically sound, and do the data support the conclusions?

Reviewer #1: Yes

Reviewer #2: Yes

Reviewer #3: Yes

2. Has the statistical analysis been performed appropriately and rigorously? 

Reviewer #1: No

Reviewer #2: Yes

Reviewer #3: Yes

3. Have the authors made all data underlying the findings in their manuscript fully available?

Reviewer #1: Yes

Reviewer #2: No

Reviewer #3: Yes

4. Is the manuscript presented in an intelligible fashion and written in standard English?

Reviewer #1: Yes

Reviewer #2: Yes

Reviewer #3: Yes

5. Review Comments to the Author

Reviewer #1: This is a well written report of a matched case-control study nested within the Antenatal Component of the PROMISE trial. The main hypothesis is that the observed difference observed in the parental trial between ZDV mono-therapy and ART could be explained by changes in progesterone and prolactin measured in the time window of 24-34 weeks of gestation. The fact that higher progesterone levels are associated with lower risk of (preterm birth (PTB) and/or low birthweight (LBW) is paradoxical so likely to be due to bias. Authors put the blame on misclassification of outcome and or exposure. Although, this cannot be ruled out, I have an alternative potential explanation for the findings. Unfortunately, because this alternative source of bias might have been introduced by a faulty study design, it cannot be removed but should be at least tested and discussed.

1. One of the key rules in case-control studies is that matching should be done only on potential confounders and not on mediators or colliders. There is no argument that age and country are potential confounders. However, it can be argued that gestational age is a proxy for the duration of exposure to treatment (this seems to be supported by data in Figure 1A showing that the higher the age the higher was the marker level). As such gestational age is a candidate for being a mediator in the relationship between treatment and outcome (see Figure, DAG below).

Under this assumption, if there is an unmeasured factor U which is a cause of both gestational age and outcome, and analysis is adjusted for the mediator, this factor becomes a confounder for the association between progesterone levels and risk of outcome (as denoted by pink arrows in the graph). Failing to control for U will bias the association between progesterone levels and risk of outcome (potentially reversing the RR from <1, which is the expected one, to >1 which is the paradoxical finding). This hypothesis could be tested using simulated scenario for collision bias such as described here https://selection-bias.herokuapp.com/.

2. Still under the assumed DAG below, the key hypothesis lends itself to formally perform a mediation analysis as progesterone/prolactin are on the causal pathway from exposure to outcome. The authors correctly evaluate first the association between type of ART and progesterone/prolactin and separately the association between progesterone/prolactin levels and risk of PTB/LBW. However, the two analyses were not put together in a formal mediation analysis showing what percentage of the total effect of treatment on risk of outcome might be mediated by progesterone/prolactin levels. Showing that the total effect is similar to the direct effect would re-inforce the hypothesis that if there are indirect pathways they go through other unmeasured factors (e.g. estradiol) and not via changes in progesterone/prolactin. Because of the observational nature of the study (a cohort of the combined arms of Promise trial), analysis should be done using a counterfactual framework (e.g. Medeff command in Stata or equivalent).

3. Whatever the reason for the bias, I would emphasize even more in the Discussion section that i) paradoxical findings are often explained by bias (could mention other examples, e.g. obesity and survival conditioning on ICU) and ii) the fundamental problem of having introduced bias at the design stage (either because of misclassification of outcome/exposure, or collider bias or both) that cannot be fixed at the analysis stage.

Minor point

First paragraph of the statistical analysis speaks about trends in progesterone/prolactin by gestational age. This could be interpreted as a longitudinal analysis with repeated measurements of progesterone/prolactin per participant, while it is cross-sectional. Suggest a rewording to clarify that is a simple correlation.

See Figure in pdf attached

Reviewer #2: This is an important study with great ramifications for the current global efforts toward eliminating mother to child transmission of HIV. Please find below comments for the authors to consider.

Methods

• Notwithstanding that the methods of the parent trial have been described extensively elsewhere, I recommend that the authors include relevant dates, including periods of recruitment for this study in this current article.

Results

• I recommend that the authors provide information relating to the numbers of participants potentially eligible, examined for eligibility, confirmed eligible, included in the study. Please give information separately for cases and controls.

• I recommend that the authors provide important characteristics of study participants. For example, socio-demographic and clinical characteristics. Information on exposures and potential confounders should be provided.

Discussion

• In this manuscript, the information in the first paragraph of the discussion section should be under the introduction section. Specific objectives and any prespecified hypothesis including the basis for such hypothesis should be under the introduction. Ideally, the first paragraph of the discussion should describe the key findings/results of the study.

• I recommend that the authors should clearly discuss the generalisability (external validity) of the study results.

• For the findings of this study to be useful to clinicians, public health practitioners and policy makers, it is important that the authors provide in a separate section under discussion the practical implications of their findings for policy, practice and public health.

Reviewer #3: The authors describe the results of a nested case control study of the PROMISE RCT evaluating pregnancy progesterone and prolactin levels in a subset of women with preterm and low birth weight infants appropriately matched controls. Mechanisms underlying adverse birth outcomes are poorly understood as such this manuscript seeks to contribute towards filling this gap in the literature. In this manuscript, the authors find that the ART exposure was associated with lower levels of progesterone, however contrary to their hypothesis PTB/LBW was associated with higher levels of progesterone.

I would like to congratulate the authors on a thoughtful analysis, that is a valuable addition to the literature. I have the following minor comments:

General

- There is an interchange between the use of “women living with HIV” or “HIV-positive women” could the authors be consistent throughout

Introduction

- Perhaps the authors could include some information on the normal changes in levels of progesterone and prolactin during normal pregnancy

o Linked to this - were specimens measured between 24-34 weeks because of study protocol (when specimens were collected) or because it was the most appropriate period physiologically?

Methods

- “Briefly, pregnant women living with HIV…..were enrolled from 14 weeks gestation onwards”

o Was gestational age determined at enrolment by LMP or SFH or only at delivery (using NBS)? Is enrolment GA data not available?

o Was there an upper limit for enrolment?

- “we conducted a nested case-control study to investigate the associations between mid-trimester progesterone….”

o Is this mid trimester or mid pregnancy

- “Delivery outcomes, including infant birth weight…”

o was birthweight measured at delivery or within 5 days of life (as stated in Table 1)

- “Participants were selected from two major enrolment sites”

o Could the authors justify the inclusion of only two of the enrolment sites

Discussion

- “The New Ballard Score…..could lead to misclassification of cases and controls.”

o could the authors speculate how misclassification could have lead to the unexpected results? Was this misclassification likely to be non-differential or differential?

6. PLOS authors have the option to publish the peer review history of their article (what does this mean?). If published, this will include your full peer review and any attached files.

Reviewer #1: No

Reviewer #2: **Yes: **Dr Olumuyiwa Omonaiye

Reviewer #3: No

---

## [Author Response · Author response to Decision Letter 0]

30 Jun 2021

We have uploaded a formatted version of our Response to Reviewers, as part of the attached files. We have included the text here too; however, there are sections in which the formatting may be difficult to interpret. In those instances, we ask that the reviewers please refer to the separate document for greater detail. 

REVIEWER 1

This is a well written report of a matched case-control study nested within the Antenatal Component of the PROMISE trial. The main hypothesis is that the observed difference observed in the parental trial between ZDV mono-therapy and ART could be explained by changes in progesterone and prolactin measured in the time window of 24-34 weeks of gestation. The fact that higher progesterone levels are associated with lower risk of (preterm birth (PTB) and/or low birthweight (LBW) is paradoxical so likely to be due to bias. Authors put the blame on misclassification of outcome and or exposure. Although, this cannot be ruled out, I have an alternative potential explanation for the findings. Unfortunately, because this alternative source of bias might have been introduced by a faulty study design, it cannot be removed but should be at least tested and discussed.

1. One of the key rules in case-control studies is that matching should be done only on potential confounders and not on mediators or colliders. There is no argument that age and country are potential confounders. However, it can be argued that gestational age is a proxy for the duration of exposure to treatment (this seems to be supported by data in Figure 1A showing that the higher the age the higher was the marker level). As such gestational age is a candidate for being a mediator in the relationship between treatment and outcome (see Figure, DAG below).

Under this assumption, if there is an unmeasured factor U which is a cause of both gestational age and outcome, and analysis is adjusted for the mediator, this factor becomes a confounder for the association between progesterone levels and risk of outcome (as denoted by pink arrows in the graph). Failing to control for U will bias the association between progesterone levels and risk of outcome (potentially reversing the RR from <1, which is the expected one, to >1 which is the paradoxical finding). This hypothesis could be tested using simulated scenario for collision bias such as described here https://selection-bias.herokuapp.com/.

RESPONSE: We appreciate this comment and we agree that the careful selection of matching factors is essential. However, we respectfully disagree about the relationships described in the attached DAG. In our trial, women were randomized and started an antiretroviral regimen after screening for the study, implying that ART cannot be a cause of gestational age at the time of ART initiation. Because gestational age at the time of ART is a baseline factor in this randomized study, it seems more reasonable to place it as an upstream factor with no ART relationship. Furthermore, the time of initiation alone cannot serve as a reliable proxy for duration of the assigned regimen, since it is wholly dependent on the time of delivery, which as the parent PROMISE trial has shown varies by study arm. For these reasons, we do not agree with the proposed mediator relationship between “ART” and “gestational age at time of ART” and do not believe that this specific example of unmeasured confounding is valid in our trial scenario. 

Having said that, we are in full agreement with the reviewer’s broader comment about unmeasured confounding. As with any study, unmeasured confounding can lead to unanticipated findings, including the seemingly paradoxical results we report here. Because this is a secondary analysis—and because the primary trial was not designed with this question in mind—we do not have the necessary data to rigorously conduct the hypothesis testing suggested above. We have included text in the Discussion that explicitly acknowledges this limitation (lines 296-299). 

2. Still under the assumed DAG below, the key hypothesis lends itself to formally perform a mediation analysis as progesterone/prolactin are on the causal pathway from exposure to outcome. The authors correctly evaluate first the association between type of ART and progesterone/prolactin and separately the association between progesterone/prolactin levels and risk of PTB/LBW. However, the two analyses were not put together in a formal mediation analysis showing what percentage of the total effect of treatment on risk of outcome might be mediated by progesterone/prolactin levels. Showing that the total effect is similar to the direct effect would re-inforce the hypothesis that if there are indirect pathways they go through other unmeasured factors (e.g. estradiol) and not via changes in progesterone/prolactin. Because of the observational nature of the study (a cohort of the combined arms of Promise trial), analysis should be done using a counterfactual framework (e.g. Medeff command in Stata or equivalent).

RESPONSE: We agree that a formal mediation analysis would strengthen our assertion in the Discussion about indirect pathways. Indeed, in an analysis using the procedures recommended above, we found that the progesterone levels contributed very little to the observed effect (see below). 

Causal Mediation Analysis

Quasi-Bayesian Confidence Intervals

 Estimate 95% CI Lower 95% CI Upper p-value 

ACME (control) -0.0105 -0.0227 0.00 0.024 * 

ACME (treated) -0.0172 -0.0366 0.00 0.024 * 

ADE (control) 0.1821 0.1305 0.23 <2e-16 ***

ADE (treated) 0.1754 0.1259 0.23 <2e-16 ***

Total Effect 0.1649 0.1120 0.22 <2e-16 ***

Prop. Mediated (control) -0.0614 -0.1576 -0.01 0.024 * 

Prop. Mediated (treated) -0.1023 -0.2461 -0.01 0.024 * 

ACME (average) -0.0138 -0.0296 0.00 0.024 * 

ADE (average) 0.1787 0.1279 0.23 <2e-16 ***

Prop. Mediated (average) -0.0818 -0.2043 -0.01 0.024 * 

---

Signif. codes: 0 '***' 0.001 '**' 0.01 '*' 0.05 '.' 0.1 ' ' 1

Sample Size Used: 298

One highlight of the analysis is that the direct effect (0.1787) is larger than the total effect of the ART arm (0.1649). This is also why the signs for the mediated effect is negative (-0.0296). Stated another way, progesterone as mediator is “blunting” the effect of ART. One could say that, under the model, there is another pathway from ART to prematurity that is larger than the pathway through progesterone.

While we agree this type of analysis would strengthen the case for indirect pathways, after thorough discussions among team members, we ultimately decided not to include it formally in this paper. Such an analysis would require greater depth than is possible in the current report and we are exploring a possible separate manuscript to address these findings. 

In addition, many factors may have led to our somewhat paradoxical findings, including misclassification, bias, and unmeasured confounding. Without a better accounting of those factors, we were concerned that inclusion of such a mediation analysis would place undue weight on our primary findings and that this could lead to misinterpretations of our results in the broader context. 

3. Whatever the reason for the bias, I would emphasize even more in the Discussion section that i) paradoxical findings are often explained by bias (could mention other examples, e.g. obesity and survival conditioning on ICU) and ii) the fundamental problem of having introduced bias at the design stage (either because of misclassification of outcome/exposure, or collider bias or both) that cannot be fixed at the analysis stage.

RESPONSE: We fully agree. We have included this as a potential explanation in the Discussion (lines 269-271). 

Minor point

First paragraph of the statistical analysis speaks about trends in progesterone/prolactin by gestational age. This could be interpreted as a longitudinal analysis with repeated measurements of progesterone/prolactin per participant, while it is cross-sectional. Suggest a rewording to clarify that is a simple correlation.

RESPONSE: We have edited the text to address this issue (lines 132-134): “For each participant, we measured progesterone and prolactin at a single time point. In our analysis, we described the distribution of these measurements, using linear regression to illustrate trends over time for the study population.” 

REVIEWER 2 

This is an important study with great ramifications for the current global efforts toward eliminating mother to child transmission of HIV. Please find below comments for the authors to consider.

RESPONSE: Thank you for the positive comments. We address each of the comments individually below.

Methods

1. Notwithstanding that the methods of the parent trial have been described extensively elsewhere, I recommend that the authors include relevant dates, including periods of recruitment for this study in this current article.

RESPONSE: The period of recruitment was included in the original manuscript (now line 169). This substudy did not recruit or enroll women separately. Instead, we analyzed stored specimens from participants who met eligibility criteria as cases and selected controls.

Results

2. I recommend that the authors provide information relating to the numbers of participants potentially eligible, examined for eligibility, confirmed eligible, included in the study. Please give information separately for cases and controls.

RESPONSE: These details have been included in lines 169-174. 

3. I recommend that the authors provide important characteristics of study participants. For example, socio-demographic and clinical characteristics. Information on exposures and potential confounders should be provided.

RESPONSE: These data were included in Table 1 of the original submission. This included key maternal and newborn characteristics, compared by case and control groups. 

Discussion

4. In this manuscript, the information in the first paragraph of the discussion section should be under the introduction section. Specific objectives and any prespecified hypothesis including the basis for such hypothesis should be under the introduction. Ideally, the first paragraph of the discussion should describe the key findings/results of the study.

RESPONSE: This first paragraph has been revised to address these critiques. We still describe our central hypothesis, but in a way that provides context to the summary of key results that follow. 

5. I recommend that the authors should clearly discuss the generalisability (external validity) of the study results.

RESPONSE: We comment on the external validity of our findings in lines 292-296. While this study was only conducted at two of the PROMISE sites, the findings are likely generalizable to the full study population, given the standardized approach to clinical management and the biological nature of the primary exposures (i.e., progesterone, prolactin). 

6. For the findings of this study to be useful to clinicians, public health practitioners and policy makers, it is important that the authors provide in a separate section under discussion the practical implications of their findings for policy, practice and public health.

RESPONSE: This analysis explores the causal pathways between HIV, ART, and adverse birth outcomes. The underlying biology of these mechanisms is important and, if properly elucidated, could lead to important interventions to prevent preterm birth. However, such developments are downstream of the current research. At this stage, our somewhat paradoxical findings do not have many practical implications for policy and practice. Instead, we offer recommendations to other clinical and translational investigators about how these findings might shape future work.

REVIEWER 3 

The authors describe the results of a nested case control study of the PROMISE RCT evaluating pregnancy progesterone and prolactin levels in a subset of women with preterm and low birth weight infants appropriately matched controls. Mechanisms underlying adverse birth outcomes are poorly understood as such this manuscript seeks to contribute towards filling this gap in the literature. In this manuscript, the authors find that the ART exposure was associated with lower levels of progesterone, however contrary to their hypothesis PTB/LBW was associated with higher levels of progesterone. I would like to congratulate the authors on a thoughtful analysis, that is a valuable addition to the literature. I have the following minor comments:

RESPONSE: Thank you so much for the positive comments. We have addressed each of the critiques below.

General

1. There is an interchange between the use of “women living with HIV” or “HIV-positive women” could the authors be consistent throughout

RESPONSE: For consistency, we have changed this to “HIV-positive women” throughout the body of the text. 

Introduction

2. Perhaps the authors could include some information on the normal changes in levels of progesterone and prolactin during normal pregnancy

RESPONSE: We agree with this suggestion but, given the flow of the paper, we found it difficult to include these details in the Introduction. However, we have included this detail in the Methods, along with supporting citations (lines 122-123).

3. Linked to this - were specimens measured between 24-34 weeks because of study protocol (when specimens were collected) or because it was the most appropriate period physiologically?

RESPONSE: We originally targeted the early third trimester (i.e., 28 to 30 weeks gestation), reasoning that it would be late enough to be proximal to time of most deliveries while being early enough for intervention (if a relationship were to be found). Due to the distribution of specimens (which was timed from study entry rather than specific gestational ages), we expanded this window to 24 to 34 weeks. This is described in further detail in lines 122-128. 

Methods

4. “Briefly, pregnant women living with HIV…..were enrolled from 14 weeks gestation onwards.” Was gestational age determined at enrolment by LMP or SFH or only at delivery (using NBS)? Is enrolment GA data not available? Was there an upper limit for enrolment?

RESPONSE: At time of enrollment, gestational age was determined according to local obstetrical practices, typically measured by last menstrual period and physical exam. The earliest a participant could be enrolled was 14 weeks gestation; because of the limited availability of obstetrical ultrasound, however, most enrollments occurred later in the second trimester. There was no upper limit to enrollment.

While local practices were used to determine gestational age for eligibility, the gestational age used in our analyses was based on newborn examination. Therefore, the timing of enrollment and timing of specimen collection noted in our results are based on that back-calculation. This has been used across the different primary and secondary analyses of the PROMISE trial, and we use that same approach here for the sake of consistency and comparability.

5. “we conducted a nested case-control study to investigate the associations between mid-trimester progesterone….” Is this mid trimester or mid pregnancy?

RESPONSE: Thank you for pointing this out. We agree with the characterization of “mid-pregnancy” and have made this change. 

6. “Delivery outcomes, including infant birth weight…” Was birthweight measured at delivery or within 5 days of life (as stated in Table 1)?

RESPONSE: We document weight at birth for those with institutional deliveries; however, some births occurred outside the facility. For those newborns, birth weights were collected as soon as possible afterwards (and within five days). To minimize confusion around this point, we have edited this sentence to read: “Delivery and early neonatal outcomes were collected on all infants born in the study” (line 97). 

7. “Participants were selected from two major enrolment sites.” Could the authors justify the inclusion of only two of the enrolment sites

RESPONSE: Because of funding constraints, we were only able to conduct the study in two sites. Both were already linked to the laboratory at Johns Hopkins that performed the progesterone and prolactin testing. We have revised the highlighted sentence to minimize confusion. Lines 104-106 now read: “This substudy analyzed stored specimens and clinical data from two major enrollment sites for the PROMISE study: Makerere University–Johns Hopkins University Research Collaboration (Kampala, Uganda) and College of Medicine-Johns Hopkins Research Project (Blantyre, Malawi).” We also note this as a possible limitation in lines 292-296.

Discussion

8. “The New Ballard Score…..could lead to misclassification of cases and controls.” Could the authors speculate how misclassification could have lead to the unexpected results? Was this misclassification likely to be non-differential or differential?

RESPONSE: In secondary analysis of the PROMISE trial, when compared to the gold standard of obstetric ultrasound, we found that the performance of the New Ballard Score varied at different gestational age thresholds. If this misclassification were to be more likely among the cases or the controls (which seem plausible, since the groups were also determined according to gestational age, or proxies thereof), then it could affect our results (lines 258-262).

JOURNAL REQUIREMENTS

and

RESPONSE: We have re-formatted the paper to comply with PLOS ONE style requirements.

2. In your Methods section, please provide additional information about the participant recruitment method and the demographic details of your participants. Please ensure you have provided sufficient details to replicate the analyses such as: a) a description of how participants were recruited, and b) descriptions of where participants were recruited and where the research took place (hospital/site name).

RESPONSE: We have added details about how cases and controls were selected to be part of the study. We have also included the site names in the Methods to provide greater clarity as to where the participants were recruited. 

3. Please ensure you have included the registration number for the clinical trial referenced in the manuscript.

RESPONSE: These have been added in line 84.

4. Please ensure you have discussed any potential limitations of your study in the Discussion, including study design, sample size and/or potential confounders.

RESPONSE: Although limitations had been discussed previously, we provide a clearer accounting of these issues in the revised manuscript. These can be found in lines 292-305.

5. Please provide additional details regarding participant consent. In the ethics statement in the Methods and online submission information, please ensure that you have specified (1) whether consent was informed and (2) what type you obtained (for instance, written or verbal, and if verbal, how it was documented and witnessed). If your study included minors, state whether you obtained consent from parents or guardians. If the need for consent was waived by the ethics committee, please include this information.

RESPONSE: An ethics statement section has been included in lines 161-167 of the manuscript. This information has also been entered into the online submission portal. 

6. We note that you have indicated that data from this study are available upon request. PLOS only allows data to be available upon request if there are legal or ethical restrictions on sharing data publicly. For information on unacceptable data access restrictions, please see http://journals.plos.org/plosone/s/data-availability#loc-unacceptable-data-access-restrictions.

RESPONSE: The IMPAACT network has established procedures related to data sharing, which are publicly available (https://www.impaactnetwork.org/resources/manual-procedures). As a study funded by this network, we are bound to these policies. Having said that, we have modified the language previously provided, so that it better explains the process. This text was used in another PROMISE analysis published in PLOS ONE in January 2020. As such, we hope it meets the journal’s requirements about this important issue: 

“Due to ethical restrictions in the study’s informed consent documents and in the IMPAACT Network’s approved human subjects protection plan, study data are available upon request from sdac.data@sdac.harvard.edu with the written agreement of the International Maternal Pediatric Adolescent AIDS Clinical Trials (IMPAACT) network. Data are also available to all interested researchers upon request to the IMPAACT Statistical and Data Management Center’s data access committee (email address: sdac.data@fstrf.org); this committee reviews and responds to requests for data, obtains necessary approvals from IMPAACT leadership and the NIH, arranges for signature of a Data Use Agreement, and sends the requested data.”

7. Thank you for stating the following in the Financial Disclosure section:

[Overall support for the International Maternal Pediatric Adolescent AIDS Clinical Trials Network (IMPAACT) was provided by the National Institute of Allergy and Infectious Diseases (NIAID) with co-funding from the Eunice Kennedy Shriver National Institute of Child Health and Human Development (NICHD) and the National Institute of Mental Health (NIMH), all components of the National Institutes of Health, under Award Numbers UM1AI068632 (IMPAACT LOC), UM1AI068616 (IMPAACT SDMC) and UM1AI106716 (IMPAACT LC), and by NICHD contract number HHSN275201800001I. Additional investigator support was provided by NIAID (BHC, K24AI120796) and the Fogarty International Center (JTP, K01TW010857). The study products in the PROMISE trial were provided free of charge by Abbott, Gilead Sciences, Boehringer Ingelheim, and GlaxoSmithKline. The content is solely the responsibility of the authors and does not necessarily represent the official views of the National Institutes of Health. For this substudy, the funders had no role in study design, data collection and analysis, decision to publish, or preparation of the manuscript.]. 

We note that you received funding from a commercial source: Abbott, Gilead Sciences, Boehringer Ingelheim, and GlaxoSmithKline

RESPONSE: We have included text for Competing Interests Statement in the cover letter, which reads:

“The study products in the PROMISE trial were provided free of charge by Abbott, Gilead Sciences, Boehringer Ingelheim, and GlaxoSmithKline. This does not alter our adherence to PLOS ONE policies on sharing data and materials. Funders had no role in study design, data collection and analysis, decision to publish, or preparation of the manuscript. The authors have declared that no competing risks exist.”

If additional information is needed, please let us know.

RESPONSE: This has been added according to PLOS ONE guidance.

---

## [Decision Letter · Decision Letter 1]

30 Aug 2022

PONE-D-20-39706R1Progesterone and prolactin levels in pregnant women living with HIV who delivered preterm and low birthweight infants: a nested case-control studyPLOS ONE

Dear Dr. Chi,

Thank you for submitting your revised manuscript to PLOS ONE - please accept my apologies for the extended delay in processing your manuscript while we sought input from an additional statistical expert. After careful consideration, we feel that it has merit but does not fully meet PLOS ONE’s publication criteria as it currently stands. Therefore, we invite you to submit a revised version of the manuscript that addresses the points raised during the review process.

 Your manuscript has been reassessed by a reviewer from the previous round, and, as mentioned above, we have obtained an additional statistical review. As you will see from the full reports below comments, the statistical review has highlighted some remaining concerns with the choice and execution of the analyses - these should be addressed before your manuscript can be deemed suitable for publication.

We look forward to receiving your revised manuscript.

Kind regards,

Dr Joseph Donlan

Senior Editor

PLOS ONE

Journal Requirements:

Reviewers' comments:

Reviewer's Responses to Questions

**Comments to the Author**

1. If the authors have adequately addressed your comments raised in a previous round of review and you feel that this manuscript is now acceptable for publication, you may indicate that here to bypass the “Comments to the Author” section, enter your conflict of interest statement in the “Confidential to Editor” section, and submit your "Accept" recommendation.

Reviewer #2: All comments have been addressed

Reviewer #4: (No Response)

2. Is the manuscript technically sound, and do the data support the conclusions?

Reviewer #2: Yes

Reviewer #4: Partly

3. Has the statistical analysis been performed appropriately and rigorously? 

Reviewer #2: I Don't Know

Reviewer #4: No

4. Have the authors made all data underlying the findings in their manuscript fully available?

Reviewer #2: Yes

Reviewer #4: No

5. Is the manuscript presented in an intelligible fashion and written in standard English?

Reviewer #2: Yes

Reviewer #4: Yes

6. Review Comments to the Author

Reviewer #2: The authors have addressed all my comments. Hence, I have no further comments. I recommend the article to be accepted for publication.

Reviewer #4: General comments:

The authors present analyses of a subset of participants from a randomized controlled trial. I have a couple first some specific questions and comments about the analytic methods presented in the paper.

My biggest question was about the choice of a case control design and conditional logistic regression. Case-control designs are great in certain situations, but tend to have lots of potential biases. Plus, in my experiences, conditional logistic regression is limiting and has some quirks that sometimes lead to odd performance. I'm wondering if utilizing propensity scores or disease risk scores would be helpful. Since you have a continuous exposure, maybe using continuous propensity scores (https://doi.org/10.1515/jci-2014-0022) maybe incorporating stratification (https://doi.org/10.1002/sim.8835).

Specific comments:

1. (lines 122-129) I can understand why the timing of samples is important and appreciate that the authors have taken care to perform an adjustment. My issue here is that that the uncertainty introduced from the adjustment method is not incorporated into the modeling results. I consider what you've done similar to imputation with a single point and usually methods like multiple imputation are considered superior because they incorporate the uncertainty into the model. That said, I would think a Bayesian analysis would be superior here since the authors could place a distribution on the points that were converted. Based on the response to comment 2 from reviewer 1, the authors may have the experience to implement this.

2. (line 133, lines 187-190, figure 1) Usually it's a strong assumption that a biomarker concentration increases linearly over time, though this may be true for progesterone and prolactin. I suggest including some information and justification on why this is a reasonable assumption in line 133. For lines 187-190, there are no confidence bands on the plots in figure 1 so it's not possible to tell if the lines really suggest a difference. In addition, I strongly encourage the authors to model these non-linearly since it's possible there's a nonlinear increase, especially for cases.

3. (lines 144-147) Given that the participants were ascertained as part of a randomized trial, I doubt the participants of the main trial represent any sort of meaningful group. Thus, it's not clear to me why weights are needed to "better replicate the full PROMISE study population" because "of the under-sampling of controls through our matching process." To me, that's an indication that matching is probably not the best approach to achieving a better estimate of the association.

4. (Table 1) Significance testing for baseline imbalance in randomized trials has been regarded as unnecessary (see Altman, https://doi.org/10.2307/2987510; Senn, https://doi.org/10.1002/sim.4780131703). This extends to observational analyses. I recommend removing the significance testing entirely from table 1. If an evaluation of difference is still desired, a standardized difference is better (see Austin, https://doi.org/10.1080/03610910902859574).

5. (fig 1) If you can, please include the definitions for the points in the figure legend.

7. PLOS authors have the option to publish the peer review history of their article (what does this mean?). If published, this will include your full peer review and any attached files.

Reviewer #2: **Yes: **Olumuyiwa Omonaiye

Reviewer #4: No

---

## [Author Response · Author response to Decision Letter 1]

21 Oct 2022

NOTE: We have uploaded a response as part of the manuscript files. We recognize that some of the formatting for the tables below may be difficult to read. If that is the case, we ask that editors/reviewers refer to the uploaded document for that information.

General comments

My biggest question was about the choice of a case control design and conditional logistic regression. Case-control designs are great in certain situations, but tend to have lots of potential biases. Plus, in my experiences, conditional logistic regression is limiting and has some quirks that sometimes lead to odd performance. I'm wondering if utilizing propensity scores or disease risk scores would be helpful. Since you have a continuous exposure, maybe using continuous propensity scores (https://doi.org/10.1515/jci-2014-0022) maybe incorporating stratification (https://doi.org/10.1002/sim.8835).

RESPONSE: We thank the reviewer for the question about the study design and suggestions. A case-control design was selected because we did not have sufficient funding to assay progesterone and prolactin on the over 3000 pregnant people who participated in the PROMISE study. In addition, we wanted to have a sufficient number of events to increase precision and allow the possibility of adjusting for a wide range of control variables. We agree that case-control designs have potential biases, but we believe that many such biases were limited in our study because the study was nested within the PROMISE study. This included matching variables and case status measured on participants who were not part of this specific analysis. Although we agree that conditional logistic regression—the standard analysis approach for case-control studies—has its drawbacks, this analysis strategy was sufficient and reliable enough to compute the odds ratios of interest in our study. 

Nevertheless, we appreciate the suggestion of propensity scores as an alternative analysis strategy. The propensity score included the matching variables, but we used a discrete propensity model since the analysis strategy for the outcome model compared groups at different percentile levels. As can be seen in the two tables below, adjustment by propensity score tended to make the odd ratios more extreme. 

This adjunct analysis reassures us about the direction of the results presented in the manuscript. As we do not have a specific reason why the propensity weighted analysis would be preferable than the conditional logistic regression, we prefer to use the results from conditional logistic regression, which appears to be more conservative. 

 Unadjusted Adjusted 

Outcome Measure Covariate Level Odds Ratio (95% CI) p-value Odds Ratio (95% CI) p-value 

PTB or LBW progesterone < 10th percentile Ref Ref 

 ≥ 10th percentile 2.61 (1.84,3.70) <.0001 2.86 (1.84,4.44)

 <.0001 

 < 25th percentile Ref Ref 

 ≥ 25th percentile 1.68 (1.21,2.32) 0.0019 2.12 (1.43,3.13)

 0.0002 

 prolactin < 10th percentile Ref Ref 

 ≥ 10th percentile 1.60 (1.16,2.22) 0.0045 2.08 (1.35,3.22)

 0.0010 

 < 25th percentile Ref Ref 

 ≥ 25th percentile 1.49 (1.08,2.06) 0.0159 1.53 (1.04,2.25) 0.0291 

 Unadjusted Adjusted for Arm and Possible Confounders

Outcome Measure Covariate Level Odds Ratio (95% CI) p-value Odds Ratio (95% CI) p-value 

PTB progesterone < 10th percentile Ref Ref 

 ≥ 10th percentile 5.70 (3.64,8.92)

 <.0001 9.15 (5.15,16.3) <.0001 

 < 25th percentile Ref Ref 

 ≥ 25th percentile 2.12 (1.47,3.06)

 <.0001 3.08 (1.97,4.80) <.0001 

 prolactin < 10th percentile Ref Ref 

 ≥ 10th percentile 1.84 (1.27,2.65)

 0.0012 2.10 (1.31,3.35) 0.0019 

 < 25th percentile Ref Ref 

 ≥ 25th percentile 2.02 (1.40,2.91) 0.0002 1.98 (1.29,3.02) 0.0016 

Specific comments:

1. (lines 122-129) I can understand why the timing of samples is important and appreciate that the authors have taken care to perform an adjustment. My issue here is that that the uncertainty introduced from the adjustment method is not incorporated into the modeling results. I consider what you've done similar to imputation with a single point and usually methods like multiple imputation are considered superior because they incorporate the uncertainty into the model. That said, I would think a Bayesian analysis would be superior here since the authors could place a distribution on the points that were converted. Based on the response to comment 2 from reviewer 1, the authors may have the experience to implement this.

RESPONSE: We thank the reviewer for the comment and the interesting perspective on the how we categorized progesterone and prolactin. While we agree that incorporating uncertainty in the cutoff used is important, the amount of additional uncertainty is minimal because there is uncertainty only in participants who have values around the cutoff. For example, a participant with a value in the 80th percentile is very unlikely to, at a population level, have a value < 10th or 25th percentile. When incorporating uncertainty into the cut-offs, only a handful of participants change comparison groups. To see this directly, we utilized multiple imputation using quantile regression and a polynomial predictors for gestational age. We then imputed the population-level percentile cutoff and averaged the results using the standard multiple imputation combination rules. One challenge we encountered in implementing this strategy is that the category below the 10th percentile contains so few participants that the imputation strategy often failed due to model instability. For this reason, we included the results for the 25th percentile below. As can be seen in the results below, the results were similar due to the reasons noted above. 

While these findings were informative, no changes were made to the manuscript. The results were similar and we were unable to provide reliable estimates for the analysis that compares groups cut at the 10th percentile. 

 Unadjusted Adjusted for Possible Confounders

Outcome Measure ARV Arm Odds Ratio (95% CI) p-value Odds Ratio (95% CI) p-value 

progesterone ZDV Only Ref Ref 

 ART Group 1.99 (1.26,3.14) 0.0033 1.71 (1.09,2.69) 0.0200 

prolactin ZDV Only (Ref.) (Ref.) 

 ART Group 1.05 (0.68,1.63) 0.8123 1.10 (0.67,1.80) 0.7015 

2. (line 133, lines 187-190, figure 1) Usually it's a strong assumption that a biomarker concentration increases linearly over time, though this may be true for progesterone and prolactin. I suggest including some information and justification on why this is a reasonable assumption in line 133. For lines 187-190, there are no confidence bands on the plots in figure 1 so it's not possible to tell if the lines really suggest a difference. In addition, I strongly encourage the authors to model these non-linearly since it's possible there's a nonlinear increase, especially for cases.

RESPONSE: We agree this can be a strong assumption. The linear line was originally intended as a visual aid, but we agree that it may be misleading. We have updated the manuscript to include new plots that utilize LOESS curves and 95% confidence intervals (as suggested). 

3. (lines 144-147) Given that the participants were ascertained as part of a randomized trial, I doubt the participants of the main trial represent any sort of meaningful group. Thus, it's not clear to me why weights are needed to "better replicate the full PROMISE study population" because "of the under-sampling of controls through our matching process." To me, that's an indication that matching is probably not the best approach to achieving a better estimate of the association.

RESPONSE: Thank you for the comment and perspective. While clinical trials are often criticized for a lack of external validity, we believe these concerns also depend on the study conducted. In this case, we have no reason to believe that the participants enrolled in PROMISE, with a population defined by the study entry criteria, would have a much different risk of an infant born premature or with a low birth weight in the general population of women living with HIV who become pregnant and seek antenatal care. 

In our view, the weighted analysis is justified for several reasons. First, the parent trial has the advantage of a large sample size (over 3500 participants) across many sites in several regions. These features are unusual for a study of this nature and its findings are certainly more generalizable than smaller, single-site trials. Second, we observed higher probability of PTB-LBW in one arm (i.e., ART) over the other (i.e., antenatal ZDV). Without weights, we are concerned our findings would be biased due to conditioning on a future intercurrent event. Third, a common critique of observational studies is their lack of internal validity due to unmeasured confounding. By including selection weights to recover the original PROMISE population we retain internal validity of the randomized design. For these reasons, we believe that our efforts to increase external validity are worthwhile and serve to enhance—rather than limit—our findings. 

Regarding the reviewer’s comment about matching: this approach has a long and established history in the context of case-control studies. Despite some limitations (including those acknowledge above), it is an accepted practice and a design feature that we are unable to change at this point. 

We have not made any changes in the manuscript based on this comment. However, we are willing to add further justification in the paper itself if the reviewer considers this to be a major issue. 

4. (Table 1) Significance testing for baseline imbalance in randomized trials has been regarded as unnecessary (see Altman, https://doi.org/10.2307/2987510; Senn, https://doi.org/10.1002/sim.4780131703). This extends to observational analyses. I recommend removing the significance testing entirely from table 1. If an evaluation of difference is still desired, a standardized difference is better (see Austin, https://doi.org/10.1080/03610910902859574).

RESPONSE: Thank you for the comment. We have removed the p-values from the baseline table. 

5. (fig 1) If you can, please include the definitions for the points in the figure legend.

RESPONSE: Notations for cases (+) and controls (o) are noted in the caption.

In addition, we have replaced reference #25 in the cited bibliography. The prior reference was a conference abstract. Since our last review, the full paper has been published. We are noting this in our response, as instructed.

---

## [Decision Letter · Decision Letter 2]

8 Jan 2023

Progesterone and prolactin levels in pregnant women living with HIV who delivered preterm and low birthweight infants: a nested case-control study

PONE-D-20-39706R2

Dear Dr. Chi,

We’re pleased to inform you that your manuscript has been judged scientifically suitable for publication and will be formally accepted for publication once it meets all outstanding technical requirements.

Kind regards,

Joseph Donlan

Staff Editor

PLOS ONE

Additional Editor Comments (optional):

Reviewers' comments:

Reviewer's Responses to Questions

**Comments to the Author**

1. If the authors have adequately addressed your comments raised in a previous round of review and you feel that this manuscript is now acceptable for publication, you may indicate that here to bypass the “Comments to the Author” section, enter your conflict of interest statement in the “Confidential to Editor” section, and submit your "Accept" recommendation.

Reviewer #4: All comments have been addressed

2. Is the manuscript technically sound, and do the data support the conclusions?

Reviewer #4: (No Response)

3. Has the statistical analysis been performed appropriately and rigorously? 

Reviewer #4: (No Response)

4. Have the authors made all data underlying the findings in their manuscript fully available?

Reviewer #4: (No Response)

5. Is the manuscript presented in an intelligible fashion and written in standard English?

Reviewer #4: (No Response)

6. Review Comments to the Author

Reviewer #4: (No Response)

7. PLOS authors have the option to publish the peer review history of their article (what does this mean?). If published, this will include your full peer review and any attached files.

Reviewer #4: No

---

## [Editor Report · Acceptance letter]

13 Jan 2023

PONE-D-20-39706R2 

Progesterone and prolactin levels in pregnant women living with HIV who delivered preterm and low birthweight infants: a nested case-control study 

Dear Dr. Chi:

I'm pleased to inform you that your manuscript has been deemed suitable for publication in PLOS ONE. Congratulations! Your manuscript is now with our production department. 

Kind regards, 

on behalf of

Dr Joseph Donlan 

Staff Editor

PLOS ONE